# The Outcome of Discontinuing Tyrosine Kinase Inhibitors in Advanced Sarcoma Following a Favorable Tumor Response to Antiangiogenics Therapy

**DOI:** 10.3390/jcm12010325

**Published:** 2022-12-31

**Authors:** Zhusheng Zhang, Qiyuan Bao, Yucheng Fu, Junxiang Wen, Meng Li, Zhuochao Liu, Guoyu He, Beichen Wang, Yuhui Shen, Weibin Zhang

**Affiliations:** 1Department of Orthopedics, Ruijin Hospital, Shanghai Jiao Tong University School of Medicine, Shanghai 200025, China; 2Shanghai Key Laboratory for Bone and Joint Diseases, Shanghai Institute of Traumatology and Orthopedics, Ruijin Hospital, Shanghai Jiao Tong University School of Medicine, Shanghai 200025, China

**Keywords:** sarcoma, therapeutics, drug administration schedule, tyrosine kinase inhibitor, target therapy

## Abstract

(1) Background: The use of antiangiogenic TKIs (AA-TKIs) has recently emerged as a major paradigm shift in the treatment of advanced sarcoma. However, the feasibility of drug holidays for patients demonstrating a very favorable response remains unknown. (2) Methods: We aim to explore the outcomes of patients with advanced sarcoma who discontinued AA-TKIs after a (near-) complete remission or were long-term responders. Patients with advanced disease were included if they had bilateral or multiple lung metastases, extrapulmonary recurrence, a short disease-free interval, etc., at the initiation of AA-TKIs. (3) Results: A total of 22 patients with AA-TKI discontinuation were analyzed, with a median follow-up of 22.3 months post-discontinuation. Prior to discontinuation, there were four drug-induced complete remissions (CRs), twelve surgical CRs, and six long-term responders. Disease progression was observed in 17/22 (77.3%) patients, with a median of 4.2 months. However, since the majority were still sensitive to the original AA-TKIs and amenable to a second surgical remission, 7 out of these 17 patients achieved a second CR after disease progression and were thus considered as relapse-free post-discontinuation (pd-RFS). Therefore, the pd-RFS and post-discontinuation overall survival (pd-OS) in the last follow-up were 12/22 (54.5%) and 16/22 (72.7%), respectively. Remarkably, surgical CR and drug tapering off (versus abrupt stopping) were associated with a greater pd-RFS and pd-OS (*p* < 0.05). Furthermore, higher necrosis rates (*p* = 0.040) and lower neutrophil-to-lymphocyte ratios (NLR) (*p* = 0.060) before discontinuation tend to have a better pd-RFS. (4) Conclusions: Our results suggest that AA-TKI discontinuation with a taper-off strategy might be safe and feasible in highly selected patients with advanced sarcoma. Surgical CR, NLR, and tumor necrosis rates before discontinuation were potential biomarkers for AA-TKI withdrawal.

## 1. Introduction

The use of tyrosine kinase inhibitors (TKIs) targeting tumor angiogenesis has emerged as a major paradigm shift in the treatment of recurrent or advanced sarcoma in recent years [1,2]. According to previous clinical trials, bone sarcoma demonstrated an objective response rate (ORR) of 8–22% to novel antiangiogenic TKIs (AA-TKIs), such as regorafenib [2], cabozantinib [3], and apatinib [4], while soft tissue sarcoma demonstrated an ORR of 5–13% to regorafenib [5], pazopanib [6], and anlotinib [7]. Moreover, antiangiogenics in combination with chemotherapy might further increase such anti-tumor activity, with an ORR of 24–29% in advanced sarcoma [8,9]. For long-lasting responders to these TKIs, it is common for the patients to interrupt the TKIs due to toxicity, noncompliance, or considering their disease fully controlled. However, to our knowledge, the feasibility of discontinuing TKIs in sarcoma responders has only been explored in gastrointestinal stromal tumor (GIST) patients with imatinib treatment [10,11]. Whether a drug holiday from AA-TKIs could be considered in sarcoma patients still remains unknown. Therefore, in this report, we aim to explore the safety and outcome of a drug holiday or discontinuation of AA-TKIs in patients with advanced sarcoma who demonstrate a (near-)complete remission or are long-term responders (duration of response of >6 months) [12,13] to the therapy.

## 2. Materials and Methods

The medical records of advanced bone and soft tissue sarcoma patients treated with AA-TKIs in our institution from 1 March 2017 to 30 December 2020 were reviewed. Only patients who were unlikely to be cured at the initiation of AA-TKIs were included. The definition of “unlikely-to-be-cured” was based on the previous literature (the era before the use of AA-TKIs) demonstrating that less than 5~10% patients could be long-term tumor-free, even after metastasectomy, if exhibiting the following risk factors: (a) bilateral pulmonary recurrence [14]; (b) pulmonary metastasis with local recurrence [15] or extrapulmonary lesions [16]; (c) a disease-free interval (DFI) of shorter than 18 months [14]; (d) the tumor was inoperable or incompletely resected (cytoreductive resection) [17]; and (e) multiple lesions (>5 metastases) [17,18]. Patients with one or more of the aforementioned risk factors were evaluated for their tumor status at the time of drug discontinuation. We only included cases of advanced sarcoma which demonstrated a favorable response, defined as (near-)complete remission (either by drug induction or surgical removal) or long-term responders (with a duration of response of >6 months [12]), according to the Response Evaluation Criteria in Solid Tumors (RECIST) 1.1 [19]. A temporary interruption and re-introduction of the drugs due to adverse effects or surgical procedures was not considered as drug discontinuation in our study.

Twenty-two patients with a favorable response were included in the analysis. All patients were informed about the potential outcome before the attempt to discontinue the AA-TKIs by the attending physicians, who balanced the risk of tumor progression and the patients’ demands or preferences for a drug holiday. In our study, discontinuation of AA-TKIs in this group of patients was mostly discouraged but considered relatively acceptable only in patients who completed at least 12 months of regimens with no evidence of disease (tumor-free) and had agreed to be compliant with close surveillance post-discontinuation, as well as if the patients had a strong demand for a drug holiday for any reasons (“planned”). Otherwise, it was defined as “unplanned” discontinuation if the patients insisted on discontinuing the drug against the physicians’ recommendation or without informing physicians despite a substantial risk of tumor relapse. Due to the fact that rapid tumor rebound is frequently observed following the abrupt discontinuation of AA-TKIs, we recommended a “taper-off” strategy to all patients, which included one dose level reduction for ~two cycles and a second dose level reduction for ~two cycles prior to permanent discontinuation. Tumor responses was assessed every 2–3 months of treatment, based on imaging with CT or magnetic resonance imaging (MRI) scans according to the RECIST 1.1 criteria [19], and were defined as complete response (CR), partial response (PR), stable disease (SD), and progressive disease (PD). In pulmonary lesions with substantial cavitation, we adopted a modified RECIST in which the longest diameter of cavitation was subtracted from the longest total diameter of the lesion in the same plane according to Crabb et al. [20]. Furthermore, on-treatment biopsy was available for 13 out of 22 patients and was analyzed for the tumor necrosis rate. Patients were also stratified based on whether their neutrophil-to-lymphocyte ratio (NLR) was ≥2.5 and platelet-to-lymphocyte ratio (PLR) ≥150 at the time of discontinuation [21,22].

The outcome after drug discontinuation was assessed with the following endpoints: (a) Progression-free survival post-discontinuation (pd-PFS), defined as the time between TKI discontinuation and the date of first noticeable disease recurrence, progression, or death, whichever occurred first. (b) Overall survival post discontinuation (pd-OS), defined as the time between TKI discontinuation and the date of death from any cause. Since some patients achieved a second CR upon re-challenging TKI and/or another surgical removal of the recurrence, we also measured the (c), relapse-free survival post discontinuation (pd-RFS), which was the same as pd-PFS except in cases where patients achieved a second CR of all known recurrence (i.e., tumor-free) after progression, and thus were still considered relapse-free until the recurrence re-relapsed into an irremovable state. The aggressiveness of tumor recurrence post-discontinuation was quantified using tumor doubling time (TDT) with the method proposed by Schwartz [23]. The objective response rate (ORR, CR + PR) and disease control rate (DCR, CR + PR + SD) were used to evaluate the sensitivity of a recurrence upon re-administration of AA-TKIs [24,25].

## 3. Results

### 3.1. Clinicopathological Data of the Patients

Out of 182 patients receiving AA-TKIs with advanced bone and soft tissue sarcoma, 22 patients were included according to our criteria (Figure 1). The AA-TKIs included apatinib (*n* = 14), anlotinib (*n* = 6), and pazopanib (*n* = 2). There were 10 males and 12 females, with a median age of 22.5 years (IQR, 16.3–26.5). Ten of them were diagnosed with osteosarcoma (OS), and eleven were diagnosed with other sarcomas. As of the data cutoff (1 May 2022), the median follow-up was 36.9 (IQR, 30.8–46.4) months in total and 22.3 (IQR, 16.6–28.4) months post-discontinuation. The median time off TKIs in all the patients was 15.3 months (IQR, 8.2–20.5). A total of 14 patients had an unplanned discontinuation due to the patients’ preference (*n* = 3), toxicity (*n* = 10), or comorbidity (*n* = 1). The dose reduction strategy involved abrupt discontinuation (*n* = 11) and tapering off (*n* = 11). At the time of drug discontinuation, there were 12 surgical CRs, 4 drug-induced CRs, and 6 long-term responders (Table 1). The median tumor necrosis rate was 46.67% (IQR, 25.00–90.00) in 13 of 22 cases where on-treatment biopsy was available. In patients who were long-term responders, the median duration of tumor response was 16.5 months (IQR, 13.6–21.2). No significant differences in age (*p* = 0.926), sex (*p* = 0.662), tumor quantity (*p* = 0.479), tumor burden (*p* = 0.513), or DFI (*p* = 0.694) were observed among these three groups of patients.

### 3.2. Oncological Outcomes of AA-TKI Discontinuation

The characteristics and outcomes of 22 patients following AA-TKI discontinuation are shown in Table 2 and Figure 2A. Disease progression following discontinuation was observed in 17 out of 22 patients (77.3%), including 2 cases with new metastatic sites. The median pd-PFS was 6.5 months (IQR, 2.8–13.3) in the entire cohort. A total of 5 patients were progression-free until the last follow-up, with a median time of 28.8 months post-discontinuation. In patients with disease progression following discontinuation, the median time to recurrence or progression after discontinuation was 4.2 months (IQR, 2.7–9.3). The median TDT was 53.8 days (IQR, 44.0–76.4). Since a significant proportion of recurrent tumors were still sensitive to the original AA-TKI therapy or amenable to surgical remission (including metastasectomy and radiofrequency ablation), 7 out of 17 patients achieved a second CR after disease progression and, thus, were still considered as having no evidence of disease until the last follow-up (defined as relapse-free in our study). Therefore, a total of 12 patients (54.5%) were relapse-free until the last follow-up, with a median follow-up of 25.1 months (IQR, 20.1–31.6) (Figure 2B). Therefore, the rate of pd-RFS and pd-OS was 63.6% (95% CI 46.4–87.3) and 81.8% (95% CI 67.2–99.6) at 6 months, and 63.6% (95% CI 46.4–87.3) and 77.3% (95% CI 61.6–96.9) at 12 months, respectively, in the total cohort. The median TDT of the 22 patients before taking AA-TKIs was 26.5 days (IQR, 18.8–58.1). In contrast, TDT appeared to be prolonged (53.8 days, IQR 44.0–76.4) in 17 patients with disease progression after discontinuation (*p* = 0.002).

In patients with disease progression following drug discontinuation, drug re-administration was taken in 14 cases, of which 4 were resistant to AA-TKIs while 10 had the disease controlled, including 6 CR, 1 PR, and 3 SD. Upon re-challenge of the original therapy, the ORR was similar to the original response (50.0% versus 72.7%, *p* = 0.29) while the DCR was significantly lower than the original therapeutic response (71.4% versus 100.0%, *p* = 0.017). In the last follow-up, six patients deceased, mostly due to oncological reasons, except one patient who died of infectious disease (pneumonitis and empyema).

### 3.3. Risk Factors Associated with the Outcomes Following Drug Discontinuation

We next sought to determine whether clinicopathological variables were associated with the prognoses after TKI discontinuation. In the univariate analysis, there was no significant association of pd-RFS or pd-OS with age, sex, time on TKI, concurrent radiotherapy, concurrent chemotherapy, body mass index (BMI), pathological diagnosis, DFI, tumor burden, or TKI re-administration (Table 3). Remarkably, the initial change rate of a tumor response (*p* = 0.721) had no significant association with the outcome after drug discontinuation (Figure 3). However, we found that drug withdrawal by tapering off, surgical removal of all lesions before discontinuation, and planned discontinuation were associated with improved pd-RFS and pd-OS (Table 3 and Figure 4). Furthermore, a higher tumor necrosis rate was associated with a better pd-RFS (*p* = 0.040) but not pd-OS (*p* = 0.280) (Table 3 and Appendix A). Interestingly, an NLR greater than 2.5 before discontinuation (*p* = 0.060, borderline significance) was also correlated with a worse pd-RFS in patients with AA-TKI withdrawal.

### 3.4. The Potential Benefit of Drug Taper-Off Strategy

Since AA-TKI withdrawal with a taper-off strategy is associated with a better prognosis compared with abrupt discontinuation, we next address the hypothesis that it is due to prolonged disease quiescence and/or slower tumor growth following relapse. Interestingly, the disease progression risk was significantly higher in abrupt discontinuation patients compared to taper-off patients (*p* = 0.008) (Appendix A). Furthermore, we found that the growth rate of the relapsed disease tends to be slower than that of the baseline before initiating AA-TKIs (median TDT 58.08 versus 32.23 days, borderline significance, *p* = 0.054) in taper-off patients (Appendix A). In contrast, no significant difference was found in the TDT between a baseline and relapsed tumor (*p* = 0.230) in patients with abrupt discontinuation (Appendix A). As we expected, the percentage of patients receiving a second CR (surgery or radiofrequency ablation) was also significantly higher in the taper-off compared with the abrupt withdrawal strategy (71.4% versus 20.0%, borderline significance, *p* = 0.058) (Appendix A). Therefore, we proposed that the taper-off strategy might be beneficial in terms of lowering the likelihood (quiescence) and growth rate of tumor relapse, with a higher resectability of the relapse after drug discontinuation Section 3.1.

## 4. Discussion

To our knowledge, our report might be the first to investigate the outcome of AA-TKI discontinuation in advanced sarcoma following a favorable tumor response. Conventionally, patients with advanced bone and soft tissue sarcoma are given a dismal prognosis because of the limited options of further-line chemotherapy. However, the use of AA-TKIs has become a major treatment modality in relapsed and chemorefractory sarcoma in recent years. While AA-TKI monotherapy demonstrated a favorable tumor response in some patients (5–22% ORR), such a ratio is likely to increase with the combination therapy of AA-TKIs with chemotherapy [8], and immunotherapy [26,27], and other modalities [28] are currently being explored. Practically, the AA-TKI regimens continued indefinitely or until progressive disease was advocated in the majority of patients with advanced sarcoma. However, it is unsurprising that patients with drastic or long-term tumor responses might seek a “drug holiday” due to surgical remission after tumor downstaging, chronic toxicity limiting daily living, or economic considerations. The outcome of discontinuing AA-TKIs in these patients remains yet to be explored.

We, for the first time, demonstrated that discontinuation of AA-TKIs with active surveillance was safe and feasible in highly selected individuals. Specifically, patients who achieve tumor downstaging with surgical CR after AA-TKIs, and, less preferably, patients with a drug-induced CR are the reasonable candidates for drug discontinuation after a favorable tumor response. The association of planned discontinuation with better prognosis could also be attributable to surgical remission since seven of the eight planned discontinuation patients underwent surgical CR. Conversely, patients with identifiable residual disease were associated with early relapse and poor survival despite long-term control of the disease, thus being strongly discouraged from a drug holiday. Remarkably, most of the advanced sarcoma involved in our study could be considered “unlikely to be cured” even if mastectomy were performed and benefited less from surgical treatment alone. Therefore, the optimal duration of TKI maintenance (12 months in our cohort) before discontinuation in these patients remains unknown. The fact that nearly 80% of the patients experienced tumor progression after a median of 4.2 months post-discontinuation further supports the existence of residual tumors in most of these cases, albeit invisible by standard radiological criteria. This finding is consistent with the result of non-progressing GIST, where even targeting known oncogenic mutations of KIT failed to eradicate all the residual disease and drug interruption led to early tumor progression in advanced and adjuvant setting [29,30]. However, unlike the common involvement of visceral organs in GIST, more than half of the patients (12 out of 22) were still tumor-free in the last follow-up due to the possibility of a second CR (*n* = 7), which likely contributed to the favorable overall survival of the patients in our cohort.

Remarkably, our study proposed the use of tapering off over abrupt withdrawal as a potential strategy for drug discontinuation in advanced sarcoma. Interestingly, we found that the taper-off approach is potentially associated with a prolonged control of invisible residual disease (tumor quiescence) before tumor progression as well as a reduced kinetics (longer TDT) after tumor progression. The later and slower recurrence of the disease might be advantageous for patients to recover TKI usage and seek a second surgical resection, as indicated by the higher percentage of resectability in patients with the taper-off strategy than those with abrupt discontinuation in our cohort (Appendix A). Notably, we failed to demonstrate the role of a longer initial duration of AA-TKI treatment in inducing disease quiescence, as patients with >12 months on TKIs are not associated with a better prognosis following drug discontinuation. However, such results should be interpreted with caution since the choice of cutoff might be strongly biased by the fact that most patients were advised to stop AA-TKIs after at least 12 months in our hands. Therefore, the optimal duration of AA-TKIs before drug discontinuation warrants further investigation.

There are several limitations to our study. First, due to the rarity of (near-) CR of sarcoma due to AA-TKIs, our cohort lacks an appropriate control, where no drug holiday is given to patients with a favorable response, for comparing the oncological outcome following drug discontinuation. Additionally, patient selection bias with the limited sample size could not be excluded, given the retrospective nature of our study. However, our results provide the first proof of the concept that, unlike GIST, drug discontinuation might be feasible in highly selected patients with advanced sarcoma with favorable responses. Drug tapering off, close surveillance, and the multi-disciplinary remission of early recurrence appear to be potential strategies for patients seeking AA-TKI withdrawal.

## 5. Conclusions

Our results suggest that patients with surgical CR after TKI therapy, with high tumor necrosis rates and low NLRs before discontinuation were reasonable candidates for a drug holiday, and the taper-off strategy is a potentially safe and recommended way for AA-TKI discontinuation.

## Figures and Tables

**Figure 1 jcm-12-00325-f001:**
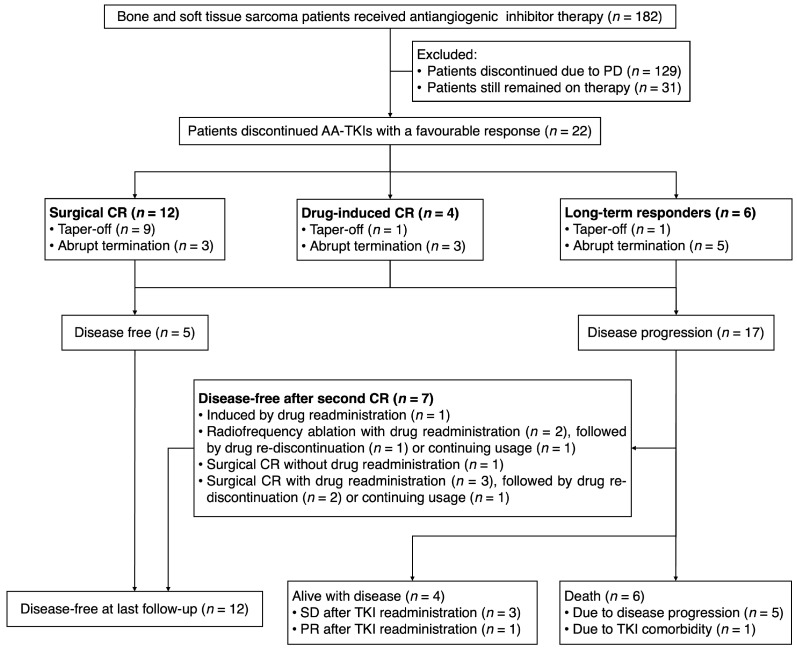
Consort flow diagram of our study. Long-term responders were defined as having a duration of response of > 6 months. PD = disease progression. CR = complete response. PR = partial response. SD = stable disease.

**Figure 2 jcm-12-00325-f002:**
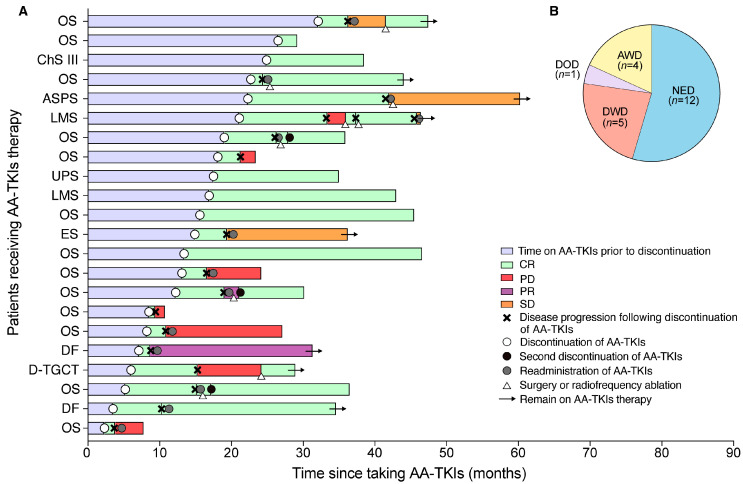
Clinical outcomes following discontinuation of TKIs. (**A**) Each bar represents one patient in this study (*n* = 22) and sarcoma subtypes are shown on Y-axis labels. Black crosses are the time-point at which radiographic disease progression occurred that was used in the analysis. Twelve patients re-administered TKI therapy after discontinuation due to disease progression. Green bars show no radiographic confirmation of tumor lesions. ASPS = alveolar soft-part sarcoma. OS = osteosarcoma. ChS III = chondrosarcoma grade III. LMS = leiomyosarcomas. UPS = undifferentiated pleomorphic sarcoma. ES = Ewing sarcoma. DF = desmoid fibromatosis. D-TGCT = diffuse-type tenosynovial giant cell tumor. CR = complete response. PD = disease progression. PR = partial response. SD = stable disease. (**B**) Pie chart shows the 22 patients’ outcomes. NED = no evidence of disease; AWD = alive with disease; DWD = died with disease; DOD = died of other disease.

**Figure 3 jcm-12-00325-f003:**
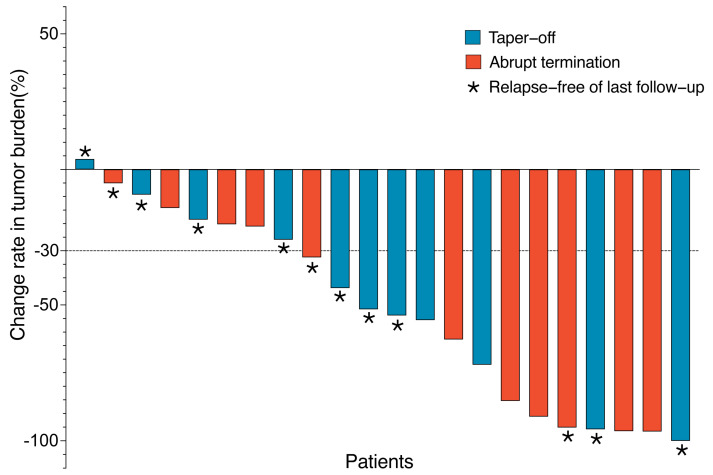
Change rate from baseline of the total tumor burden. Each bar represents one patient (*n* = 22). * These patients were disease-free in last follow-up by surgery or radiofrequency ablation. Red bars represent patients who discontinued TKI by tapering off, and the blue represents those with abrupt termination. Dashed lines indicate RECIST criteria for progressive disease (+20%) or partial response (−30%).

**Figure 4 jcm-12-00325-f004:**
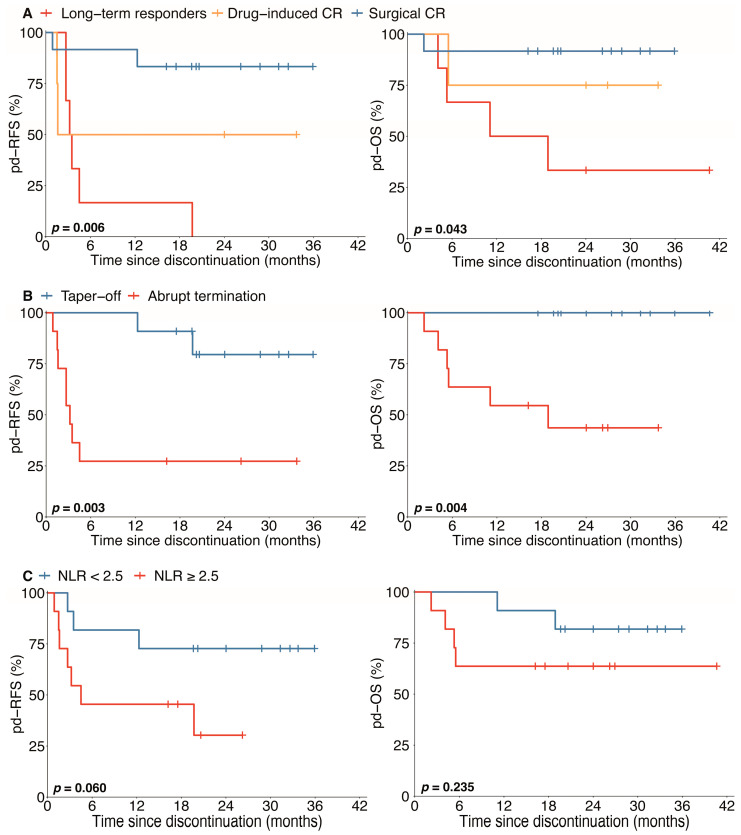
Kaplan–Meier curves for pd-RFS and pd-OS. Pd-RFS = disease-free survival since the discontinuation of TKI, Pd-OS = overall survival since the discontinuation of TKI, NLR = neutrophil-to-lymphocyte ratio, and CR = complete response. Kaplan–Meier analysis estimates show the distribution of RFS and OS for all patients with TKI discontinuation analysis (*n* = 22). Crosses indicate censored patients. Surgical CR before discontinuation (**A**) was independently associated with improved survival (*p* = 0.006 for pd-RFS and *p* = 0.043 for pd-OS). Discontinuation by tapering off (**B**) significantly improved the RFS and OS of patients with TKI therapy (*p* = 0.003 for pd-RFS and *p* = 0.004 for pd-OS). A lower NLR (**C**) might be a prognostic marker in pd-RFS (*p* = 0.060, borderline significance), while it showed no sensitivity in pd-OS (*p* = 0.235).

**Table 1 jcm-12-00325-t001:** Clinical characteristics of 22 patients before the AA-TKI discontinuation.

No.	Status before Discontinuation	Age	Gender	Pathological Diagnosis	Discontinuation Reason	Concurrent Radiotherapy	Concurrent Chemotherapy	Risk Factors *	Prior Lines of Therapy	Recurrence Sites at the Initiation of AA-TKIs
1	Surgical CR	25	M	OS	Unplanned (toxicity)	No	No	b, d	1	Lung
2	Surgical CR	16	M	OS	Planned	No	No	d	1	Lung
3	Surgical CR	48	F	LMS	Planned	No	No	a, c, e	1	Lung
4	Surgical CR	23	F	LMS	Planned	No	No	d	1	LR
5	Surgical CR	22	M	OS	Planned	No	No	a, b	2	Lung
6	Surgical CR	19	M	OS	Planned	No	No	c	2	Lung
7	Surgical CR	15	M	OS	Planned	No	No	c	2	Lung
8	Surgical CR	48	F	D-TGCT	Unplanned (preference)	Yes	Yes	a, b, d, e	0	Lung + LR
9	Surgical CR	54	F	UPS	Unplanned (preference)	No	No	a, c, e	0	Lung
10	Surgical CR	49	F	ChS III	Unplanned (toxicity)	Yes	Yes	c, d	1	LR
11	Surgical CR	17	M	OS	Unplanned (toxicity)	Yes	Yes	a, c, e	2	Lung
12	Surgical CR	9	F	OS	Planned	No	Yes	a, c, e	2	Lung
13	Drug-induced CR	9	F	OS	Planned	No	No	a, c	2	Lung
14	Drug-induced CR	18	M	OS	Unplanned (toxicity)	No	No	a, c, d, e	2	Lung
15	Drug-induced CR	43	F	DF	Unplanned (toxicity)	Yes	Yes	d	0	LR
16	Drug-induced CR	27	F	DF	Unplanned (toxicity)	Yes	Yes	d	0	LR
17	Long-term responders	23	F	ASPS	Unplanned (preference)	Yes	Yes	a, d, e	1	Lung
18	Long-term responders	19	M	ES	Unplanned (toxicity)	No	No	a, d	1	Lung
19	Long-term responders	16	F	OS	Unplanned (toxicity)	No	No	c	2	Lung
20	Long-term responders	24	M	OS	Unplanned (toxicity)	No	No	a, c	1	Lung
21	Long-term responders	25	M	OS	Unplanned (comorbidity)	No	No	a, b, d, e	1	Lung
22	Long-term responders	13	F	OS	Toxicity	Yes	Yes	a, c, d, e	2	Lung

F = female; M = male; UPS = undifferentiated pleomorphic sarcoma; ASPS = alveolar soft-part sarcoma; D-TGCT = diffuse-type tenosynovial giant cell; OS = osteosarcoma; ES = Ewing Sarcoma; LMS = leiomyosarcoma; DF = desmoid fibromatosis; ChS III = chondrosarcoma grade III; 0 = have no chemotherapy before taking TKI; 1 = have one chemotherapy regimen; 2 = have two chemotherapy regimens; LR = local recurrence; * (a) bilateral pulmonary recurrence; (b) pulmonary metastasis with local recurrence or extrapulmonary lesions; (c) DFI shorter than 18 months; (d) the tumor was inoperable or incomplete resected (cytoreduction); (e) multiple lesions (>5 metastases).

**Table 2 jcm-12-00325-t002:** Clinical and oncological data for the 22 patients’ outcomes following AA-TKI discontinuation.

No.	Time on TKI, Months (TKI)	Largest Tumor Burden (mm)	Total Tumor Burden (mm)	Dose Reduction	Outcome	Action Taken	Results	Survival Status	Follow-Up, Months
1	5.2 (a)	Lung: 52.0–2.2	52.0–2.2	Taper-off	New lesion of bone	Second surgical CR without TKI re-administration	CR	NED	36.5
2	15.6 (a)	Lung: 54.0–40.0	54.0–40.0	Taper-off	Progression-free	Active surveillance	/	NED	48.2
3	21.1 (a)	Lung: 14.0–14.0	33.3–28.4	Taper-off	Pulmonary recurrence	Surgery and radiofrequency with TKI re-administration	SD	AWD	48.5
4	16.9 (b)	LR: 104.2–58.6	104.2–58.6	Taper-off	Progression-free	Active surveillance	/	NED	45.6
5	32.1 (a)	Lung: 10.4–5.0	26.0–12.0	Taper-off	Pulmonary recurrence	TKI re-administrated	CR	NED	52.7
6	13.4 (a)	Lung: 17.0–12.0	27.0–22.0	Taper-off	Progression-free	Active surveillance	/	NED	49.3
7	19.0 (a)	Lung: 11.0–6.3	19.0–9.2	Taper-off	Pulmonary recurrence	Radiofrequency ablation	CR	NED	38.6
8	6.0 (a)	LR: 181.0–112.0	216.0–146.0	Abrupt termination	Pulmonary recurrence	Surgery	CR	NED	31.4
		Lung: 15.0–14.2							
9	17.5 (b)	Lung: 7.0–7.6	13.0–13.5	Taper-off	Progression-free	Active surveillance	/	NED	35.1
10	24.9 (b)	LR: 61.0–57.9	61.0–57.9	Abrupt termination	Progression-free	Active surveillance	/	NED	41.2
11	8.5 (a)	Lung: 19.0–15.0	19.0–15.0	Abrupt termination	Pulmonary recurrence	TKI re-administered	PD	DWD	10.7
12	12.2 (a)	Lung: 7.3–6.0	18.0–8.0	Taper-off	Pulmonary recurrence	Second surgical CR with TKI re-administration	CR	NED	32.4
13	22.7 (a)	Lung: 5.4–0.0	12.2–0.0	Taper-off	Pulmonary recurrence	Radiofrequency ablation with TKI re-administration	CR	NED	46.7
14	2.3 (a)	Lung: 28.0–5.0	140.7–5.0	Abrupt termination	Pulmonary recurrence	TKI re-administered	PD	DWD	7.8
15	7.1 (b)	LR: 146.0–5.0	146.0–5.0	Abrupt termination	Local recurrence	TKI re-administered	PR	AWD	34.0
16	3.5 (b)	LR: 103.0–5.0	103.0–5.0	Abrupt termination	Local recurrence	TKI re-administered	CR	NED	37.2
17	22.3 (a)	LR: 66–27	120.1–33.6	Taper-off	Pulmonary recurrence	TKI re-administered	SD	AWD	62.9
		Lung: 4.6–3.1							
18	14.9 (a)	Lung: 7.5–7.3	65.8–52.5	Abrupt termination	Pulmonary recurrence	TKI re-administered	PD	AWD	38.9
19	13.1 (c)	Lung: 8.4–5.0	13.4–5.0	Abrupt termination	Pulmonary recurrence and lymph nodes metastasis	TKI re-administered	PD	DWD	24.2
20	18.1 (c)	Lung: 14.0–2.2	24.9–2.2	Abrupt termination	Pulmonary recurrence	Supportive care	PD	DWD	23.3
21	26.5 (a)	Lung: 41.7–24.6	723.6–105.8	Abrupt termination	Progression-free	Supportive care	/	DOD	30.6
22	8.2 (a)	Lung: 19.7–17.1	27.5–23.6	Abrupt termination	Pulmonary recurrence	TKI re-administered	PD	DWD	27.1

Type of TKI used: a = apatinib; b = anlotinib; c = pazopanib; PD = disease progression. SD = stable disease; PR = partial response; CR = complete response; NED = no evidence of disease; AWD = alive with disease; DWD = died with disease; DOD = died of other disease.

**Table 3 jcm-12-00325-t003:** Kaplan–Meier Analysis of the independent prognostic factors for pd-RFS and pd-OS among the patients.

Variable		Number of Patients (%)	*p*-Value	*p*-Value
			(pd-RFS)	(pd-OS)
Age at first diagnosis	≥20 y	12 (54.5%)	0.574	0.293
	<20 y	10 (45.5%)		
Sex	Male	10 (45.5%)	0.546	0.177
	Female	12 (54.5%)		
Location of primary tumor	Limb	17 (77.3%)	0.726	0.153
	Axis or girdle	5 (22.7%)		
Genomic characteristics	Translocation-related	3 (13.6%)	0.739	0.253
	Genomic complex	17 (77.3%)		
	Other	2 (9.1%)		
Planned discontinuation	No	14 (63.6%)	**0.018**	**0.034**
	Yes	8 (36.4%)		
Dose reduction	Taper-off	11 (50.0%)	**0.003**	**0.004**
	Abrupt termination	11 (50.0%)		
Time on TKI	<1 years	7 (31.8%)	0.214	0.300
	>1 years	15 (68.2%)		
DFI	<18 months	16 (80.0%)	0.410	0.859
	≥18 months	4 (20.0%)		
Concurrent radiotherapy	No	16 (72.7%)	0.725	0.597
	Yes	6 (27.3%)		
Concurrent chemotherapy	No	15 (68.2%)	0.513	0.244
	Yes	7 (31.8%)		
BMI	Normal	13 (59.1%)	0.830	0.565
	Low	6 (27.3%)		
	High	3 (13.6%)		
Pathological diagnosis	Intermediate-locally aggressive	3 (13.6%)	0.739	0.286
	Malignant	19 (86.4%)		
Type of TKI	Apatinib	14 (63.6%)	0.663	0.886
	Other TKI	8 (36.4%)		
Site of recurrence	Lung only	17 (77.3%)	0.968	0.461
	Multiple sites	2 (9.1%)		
	Local recurrence	3 (13.6%)		
Change rate of the largest lesions	≥−30.0%	9 (40.9%)	0.960	0.705
	<−30.0%	13 (59.1%)		
Change rate of total lesions	≥−30.0%	8 (36.4%)	0.721	0.907
	<−30.0%	14 (63.6%)		
NLR	<2.5	11 (50.0%)	0.060	0.235
	≥2.5	11(50.0%)		
PLR	<150	13 (59.1%)	0.548	0.742
	≥150	9 (40.9%)		
Quantity of baseline lesions	<3	8 (36.4%)	0.202	0.240
	≥3	14 (63.6%)		
Bilateral pulmonary metastasis	No	6 (27.3%)	0.279	0.272
	Yes	12 (54.5%)		
	No pulmonary metastasis	4 (18.2%)		
Central pulmonary metastasis	No	10 (45.5%)	0.522	0.783
	Yes	8 (36.4%)		
	No pulmonary metastasis	4 (18.2%)		
Response with cavitation	No	16 (72.7%)	0.638	0.116
	Yes	6 (27.3%)		
TKI re-administered *	No	3 (17.6%)	0.729	0.108
	Yes	14 (82.4%)		
Tumor burden	<3.0 cm	10 (45.5%)	0.789	0.248
	≥3.0 cm	12 (54.5%)		
Tumor necrosis rate	<Median	6 (46.2%)	**0.040**	0.280
	≥Median	7 (53.8%)		
Status before discontinuation	Long-term responders	6 (27.3%)	**0.006**	**0.043**
	Drug-induced CR	4 (18.2%)		
	Surgical CR	12 (54.5%)		
Number of risk factors	<2	7 (31.8%)	0.331	0.323
	≥2	15 (68.2%)		

Log-rank test significant variables (*p*-values < 0.05) are in bold. Pd-RFS = post-discontinuation relapse free survival; pd-OS = post-discontinuation overall survival. DFI = disease-free interval, BMI = Body Mass Index, NLR = neutrophil-to-lymphocyte ratio, and PLR = platelet-to-lymphocyte ratio. * Only for 17 patients who had disease recurrence after TKI discontinuation.

## Data Availability

Not applicable.

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
