# Peer review of "The Outcome of Discontinuing Tyrosine Kinase Inhibitors in Advanced Sarcoma Following a Favorable Tumor Response to Antiangiogenics Therapy"

_jcm, 2022, doi:10.3390/jcm12010325_

Round 1

Reviewer 1 Report

This article describes, a super-select cohort of patients receiving tyrosine kinase inhibitors to treat advanced sarcoma. Those patients present good responses to the therapy. The authors want to analyze the effects on disease progression when the drug was interrupted.

Despite the small cohort (only 22) and the different diagnoses and drugs, the article points out some conclusions that are attractive.

One point I recommend further explanation:  why diffuse type tenosynovial giant cell tumor was included because the diffuse type tenosynovial giant cell tumors aren't sarcomas. 

Reviewer 2 Report

Thank you for giving me an opportunity to review this article.

I understood that this manuscript aims to clarify the feasibility and benefits of drug holidays of antiangiogenic tyrosine kinase inhibitor (AA-TKI).

I would like the authors to address some major issues about this article.

1. If the authors aim to show the feasibility of discontinuation of AA-TKI, they should show patients’ QOL. Drug discontinuation should be no meaning if no improvement of QOL is observed despite of the risk of tumor progression. I believe this is a critical point of this article.

2. Whether drug holiday is beneficial or not remains unclear after reading this article. The authors should show the difference in survival between drug holiday group and no drug holiday group.  

3. Of importance, the object (total 22 patients) includes 12 Surgical CR. As already known, metastasectomy improves overall survival, particularly in patients with lung metastases. I am aware of that removal of metastases is important rather than AA-TKI treatment. Hence, the authors should not include patients with surgical CR. This conveys a very misleading message for readers. 

Round 2

Reviewer 2 Report

The quality of manuscript is much better than the previous version. As the authors mentioned, there are some limitations in this study, and I still somewhat wonder the importance of this report because 'planned' and 'unplanned' drug holiday are mixed. I would like to ask them to add the data to separate between 'planned' and 'unplanned' drug holiday groups because the reason of drug holiday is definitively different. For example, parameter of 'planned' and 'unplanned' can be added in Table 3. 

Author Response

We greatly appreciated the reviewer's further suggestion to our manuscript, and very agree with the recommendation to include "planned" versus "unplanned" as a factor in Table3. We have accordingly updated Table3. We found that a planned discontinuation of TKIs is also significantly associated with better outcome than unplanned one. Such an association of planned discontinuation with better prognosis could probably be attributable to surgical remission, since 7 of the 8 planned discontinuation patients underwent surgical CR, in comparison to only 4 of 14 having surgery before unplanned discontinuation. In practice, we recommend a "planned" discontinuation mainly for patients with metastasectomy. Therefore, our main conclusion remain unchanged: Patients with a down-staged and resected advanced sarcoma are safe to discontinue their AA-TKIs after a duration of maintenance therapy, which is, in our hand, one year of AA-TKI. Again, we thanks the reviewer for the time and effort made for our submission.